# Near-Infrared Spectroscopy Intravascular-Ultrasound-Guided Intervention in Minimal Coronary Artery Stenosis: A Case Report

**DOI:** 10.3390/medicina58091166

**Published:** 2022-08-27

**Authors:** Hyung-Oh Kim, Jong-Shin Woo, Joan Kim, Weon Kim

**Affiliations:** 1Division of Cardiology, Sahmyook Seoul Hospital, Seoul 02500, Korea; 2Division of Cardiology, Kyung Hee University Hospital, Seoul 02447, Korea

**Keywords:** intracoronary imaging, coronary artery disease, percutaneous coronary intervention, vulnerable plaque

## Abstract

Near-infrared spectroscopy intravascular ultrasounds (NIRS-IVUSs) can identify high-risk plaque morphologies associated with future event risk. However, the usage of NIRS-IVUSs is not universal. We report a case with insignificant coronary angiography (CAG) and high-risk NIRS-IVUS findings. A 58-year-old man with exertional dyspnea was admitted for a CAG evaluation. The CAG of the patient demonstrated mild angiographic stenosis in the mid-left anterior descending artery. However, NIRS-IVUS revealed a high maximum lipid core burden index at 4 mm (MaxLCBI4mm) and an intraluminal calcific protrusion with severe luminal stenosis at the lesion. Therefore, the patient was diagnosed as stable angina, and a drug-eluting stent was implanted in the lesion. A post-stent NIRS-IVUS demonstrated improved MaxLCBI4mm and significantly improved luminal stenosis. The patient did not have any procedural complications. In the present case, a patient with insignificant CAG demonstrated multiple high-risk features on NIRS-IVUS. Therefore, a percutaneous coronary intervention was performed. The presented case highlights the utility of NIRS-IVUS in nonobstructive CAG.

## 1. Introduction

Although coronary artery angiography (CAG) has been a mainstay in the detection of the gross presence of coronary artery disease, it may underestimate the atherosclerotic burden and risks in lesions with nonobstructive luminograms. Where CAG provides two-dimensional information regarding the coronary artery, intravascular ultrasonography provides a continuum of tomographic images and, thus, yields more detailed information on vessel structure and plaque morphology [1]. Near-infrared spectroscopy intravascular ultrasounds (NIRS-IVUSs) combine spectroscopy with intravascular ultrasonography to further identify high lipid burdens and prove their clinical utility in cases with nonobstructive coronary artery angiography [2]. Here, we present the case of a patient with nonobstructive CAG in which NIRS-IVUS revealed multiple high-risk features in the lesion.

## 2. Case Report

A 58-year-old man with hypertension and end-stage renal disease (ESRD) who was on peritoneal dialysis visited an outpatient office with exertional dyspnea. He was never a smoker. His height was 163.6 cm, and his body weight was 80.1 kg. (body mass index: 30 kg/m^2^) The blood pressure of the patient was 140/90 mmHg, and the heart rate was 90 beats/min. Although the patient could be stratified as a moderate risk group based on SCORE2 scoring, ESRD was additionally considered for risk stratification [3]. Therefore, the patient was classified as high-risk and directly planned for CAG [4]. The patient showed mild stenosis in the mid-left anterior descending (LAD) artery in the exam (Figure 1).

In motion cinematography, a faint filling defect was noted in the lesion (Appendix A). The lesion was located relatively in the proximal part of LAD artery, and the contrast density within the lesion was inconsistent. The operator decided to evaluate high-risk plaque characteristics with NIRS-IVUS, for they may imply future event risk. Additional physiologic study was not performed for economic reasons. A NIRS-IVUS system was introduced beyond the lesion. Ultrasonography revealed a calcific protrusion (Figure 2a), and spectroscopy demonstrated a high lipid burden (Figure 2b and Appendix A). The maximum lipid core burden index at 4 mm (MaxLCBI_4mm_) was 462 (Figure 2c). The minimum lumen area (MLA) was 3.7 mm^2^, and the percent of area stenosis was 61.2%. This could be approximately translated into a percent diameter stenosis of 48%, which slightly did not meet significant stenosis. Despite absolute guideline cutoff in diameter, the stenosis was not enough for percutaneous coronary intervention (PCI), and the high lipid burden indicated by the MaxLCBI_4mm_ and LAD location factors could impact future cardiac event risk [2,5]. To maximally benefit the future prognosis of the patient, the operator decided to perform a PCI.

An EBU 3.5 catheter was engaged, and a Runthrough NS Hypercoat guidewire was introduced and passed down to the distal part of the LAD artery. Balloon dilation was performed with a Tytrak 2.5 mm × 20 mm semi-compliant balloon. Sequentially, a Resolute Onyx 3.5 mm × 18 mm stent was implanted. Postdilation was performed with an Accuforce 3.5 mm × 15 mm non-compliant balloon, which was inflated to 22 ATM/3.75 mm. Postprocedural NIRS-IVUS pullback demonstrated no edge dissection, no malapposition, and minimal underexpansion (Figure 3a and Appendix A). The final CAG demonstrated excellent results (Figure 3b and Appendix A).

The post-stent MaxLCBI_4mm_, MLA, and percent of area stenosis were improved to 282, 8.9 mm^2^, and 13.2%, respectively. The postprocedural changes in the quantitative profiles of the lesions are described in Table 1. The patient did not have any periprocedural complication in 24 h of monitoring in the cardiovascular care unit. However, the symptoms were not completely relieved after the procedure. At the time, the NT-proBNP was 1797 pg/mL, and an echocardiogram of the patient demonstrated no left ventricular (LV) systolic pressure, no significant valvular dysfunction, no pulmonary hypertension, and grade 1 diastolic dysfunction (LV ejection fraction 65%; only trivial tricuspid regurgitation without elevated right ventricular systolic pressure). The potential causes for the symptom could be ESRD or obesity. The patient was planned to be followed-up with by the operator and the nephrologist. He was also educated with lifestyle modification, including dietary, behavioral, and weight loss management. Already prescribed aspirin at 100 mg, olmesartan at 40 mg, and furosemide at 80 mg per day at the nephrology department, the operator additionally prescribed clopidogrel at 75 mg, a combination pill of rosuvastatin at 10 mg and ezetimibe at 10 mg, bisoprolol at 2.5 mg, and nicorandil at 10 mg per day. After the discharge, the patient was free from any major cardiac event for 6 months.

## 3. Discussion

Although the patient was admitted for exertional dyspnea, a stent implantation was performed to prevent future cardiac events, such as cardiac death or myocardial infarction. MLA ≥ 4 mm^2^ was used as a cut-off for the lower event rate [6]. A MaxLCBI_4mm_ over 400 and a 100-unit increase in NIRS-IVUS were shown to increase the risk for future events [2]. Luminal calcific protrusions are one of the major causes of acute coronary syndromes [7]. The operator identified a small MLA, a high MaxLCBI_4mm_, and calcific protrusions via NIRS-IVUS. The simultaneous presence of these high-risk factors could imply potential future cardiac events and, thus, a percutaneous coronary intervention (PCI) was performed. It should be noted that, while a greater MLA generally excludes functional ischemia, a smaller MLA does not necessarily result in it [8]. It is possible that the LAD lesion may not have been fully responsible for the exertional dyspnea of the patient; thus, the patient was followed-up with by the nephrologist.

Most studies regarding PCI in angina have shown no significant benefit in event-free survival [9], but intracoronary-imaging-based high-risk features in the LAD artery have been associated with an increased risk for future events [5]. The efficacy of preventing future events by LAD PCI is still under debate [10], and future studies involving intracoronary imaging and cardiac events are mandatory.

## 4. Conclusions

Although present consensus suggests the benefits of PCIs in stable coronary diseases are limited to symptom relieving, some prospective studies have suggested otherwise. The present case demonstrated that some high-risk lesion characteristics could be overlooked with CAG alone. Intravascular imaging techniques, such as NIRS-IVUS, may be considered in specific situations for PCIs in lesions with high-impact characteristics to minimize future event risk.

## Figures and Tables

**Figure 1 medicina-58-01166-f001:**
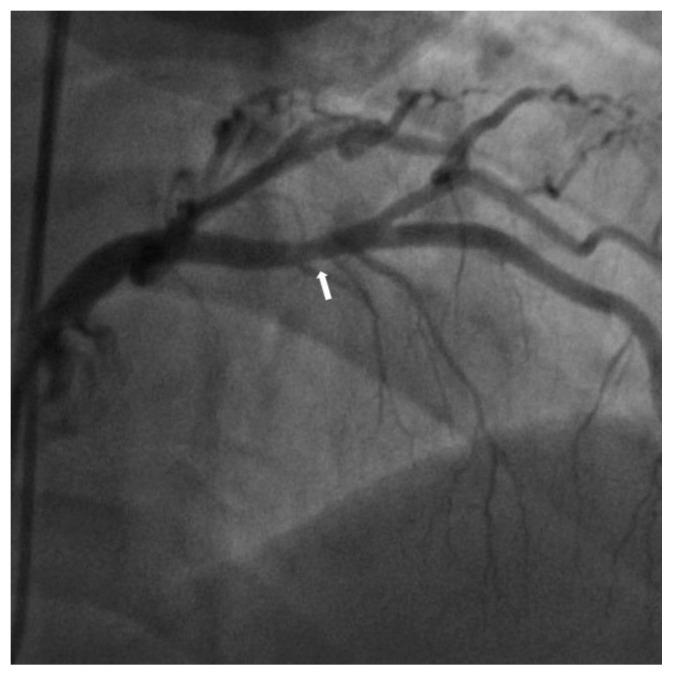
Mild stenosis (arrow) was noted in the mid-LAD artery in the RAO cranial view. RAO, right anterior oblique; CAG, coronary angiography; LAD, left anterior descending.

**Figure 2 medicina-58-01166-f002:**
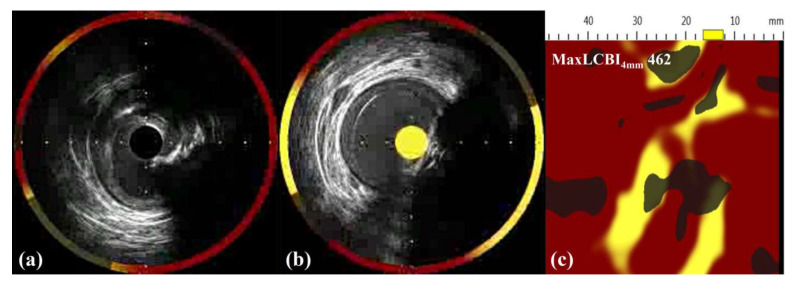
NIRS-IVUS images of the nonobstructive coronary artery lesion. NIRS-IVUS pullback revealing a calcific protrusion in the lesion area (**a**) and high lipid burden (**b**). The LCBI spread-out plot reports a MaxLCBI4mm of 462 (**c**). LCBI, lipid core burden index; MaxLCBI4mm, maximum lipid core burden index at 4 mm; NIRS-IVUS, near-infrared spectroscopy intravascular ultrasound.

**Figure 3 medicina-58-01166-f003:**
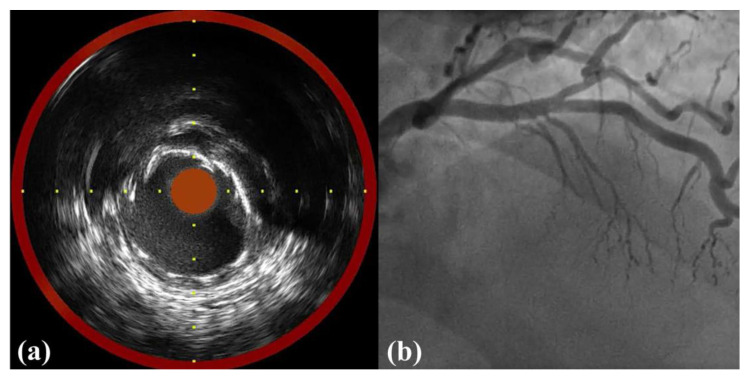
Post-procedural NIRS-IVUS and CAG images. NIRS-IVUS pullback after the non-compliant balloon demonstrated no edge dissection, no malapposition, and minimal underexpansion (**a**). Final CAG showed excellent results (**b**). CAG, coronary angiography; NIRS-IVUS, near-infrared spectroscopy intravascular ultrasound.

**Table 1 medicina-58-01166-t001:** Quantitative NIRS-IVUS profiles before and after PCI.

Profiles	Pre-PCI	Post-PCI
LCBI	211	114
maxLCBI_4mm_	462	282
Minimum dRD (mm)	2.9	3.1
Maximum dRD (mm)	3.6	3.8
dRA (mm^2^)	8.8	10.0
Minimum pRD (mm)	3.3	3.5
Maximum pRD (mm)	4.0	3.9
pRA (mm^2^)	10.3	10.5
Minimum MLD/MSD (mm)	1.8	2.9
Maximum MLD/MSD (mm)	2.5	3.7
MLA and MSA (mm^2^)	3.7	8.9
Area stenosis (%)	61.2	13.2
Lesion and stent length (mm)	17	18

dRA, distal reference area; dRD, distal reference diameter; LCBI, lipid core burden index; MaxLCBI_4mm_, maximum lipid core burden index at 4 mm; MLA, minimum lumen area; MLD, minimum lumen diameter; MSA, minimum stent area; MSD, minimum stent diameter; NIRS-IVUS, near-infrared spectroscopy intravascular ultrasonography; PCI, percutaneous coronary intervention; pRA, proximal reference area; pRD, proximal reference diameter.

## Data Availability

All data shown in this study are included in the published article.

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
