# Peer review of "Near-Infrared Spectroscopy Intravascular-Ultrasound-Guided Intervention in Minimal Coronary Artery Stenosis: A Case Report"

_medicina, 2022, doi:10.3390/medicina58091166_

Round 1

Reviewer 1 Report

Dear Authors

The "Case report" presented to me for review, entitled 'Near-Infrared Spectroscopy Intravascular Ultrasound-Guided Intervention in Minimal Coronary Artery Stenosis: a case report ", is interesting. It concerns insignificant, but nevertheless dangerous, atherosclerotic lesions in the coronary arteries.

I have a few comments:
- In the Abstract, there is no information in which artery the intervention was performed
- Was the patient on dialysis?
- The main symptom reported by the patient was dyspnea that did not resolve after the procedure? Has the patient had an echocardiographic examination? What was the valve function and contractility of the left ventricle? What was BNP or NT-pro-BNP like?
- What pharmacological treatment was used in the patient?
- In Conclusions - the first two sentences are rather not conclusions.
Best regards

Author Response

Reviewer 1

The "Case report" presented to me for review, entitled 'Near-Infrared Spectroscopy Intravascular Ultrasound-Guided Intervention in Minimal Coronary Artery Stenosis: a case report ", is interesting. It concerns insignificant, but nevertheless dangerous, atherosclerotic lesions in the coronary arteries.

I have a few comments:

- In the Abstract, there is no information in which artery the intervention was performed

We apologize for your inconvenience. We revised Abstract part as follows.

The CAG of the patient demonstrated mild angiographic stenosis in mid left anterior descending artery. (Page 1, line 14)

- Was the patient on dialysis?

We used the term ‘ESRD’ to indicate the patient was on dialysis, but to clarify further, we added description that the patient was on routine peritoneal dialysis. Case report part was revised as below.

A 58–year–old man, with hypertension and end-stage renal disease (ESRD) who was on peritoneal dialysis, visited outpatient office with exertional dyspnea. (Page 1, line 37 ~ 38)

- The main symptom reported by the patient was dyspnea that did not resolve after the procedure? Has the patient had an echocardiographic examination? What was the valve function and contractility of the left ventricle? What was BNP or NT-pro-BNP like?

We thank the Reviewer for raising several important questions related with the non-procedural aspect of the case. As we replied and revised in the above Editor’s comment, the LAD PCI was performed to benefit the prognosis of the patient, (Prati et al. 2019; Waksman et al. 2019) rather than symptom. He was never smoker, so smoking-related pulmonary disease was not one of our impressions. The NT-proBNP was1797 pg/mL at the time of PCI, but trans-thoracic echocardiogram demonstrated no left ventricular (LV) systolic / valvular dysfunction. (LV ejection fraction 65%; only trivial tricuspid regurgitation) He had non-cardiac comorbidities as ESRD and obesity. (Height 163.6cm; body weight 80.1kg; body mass index 30kg/m2) These factors might have contributed to the symptom. Nephrologist follow-up with further dialysis control and education for lifestyle modification was planned for the long-term management. We incorporated these data in the Case report and Discussion part.

He was a never smoker. His height was 163.6cm and the body weight was 80.1kg. (Body-mass-index 30kg/m2) (Page 1, line 38 ~ 40)

At the time, NT-proBNP was 1797 pg/mL and echocardiogram of the patient demonstrated no left ventricular (LV) systolic, no significant valvular dysfunction, no pulmonary hypertension, and grade 1 diastolic dysfunction. (LV ejection fraction 65%; only trivial tricuspid regurgitation without elevated right ventricular systolic pressure) The potential causes for the symptom could be ESRD and obesity. The patient was planned to be followed up by the operator and the nephrologist. He was also educated with lifestyle modification including dietary, behavioral, and weight loss management. (Page 3, line 93 ~ 100)

- What pharmacological treatment was used in the patient?

The patient was previously prescribed with aspirin 100mg qd, Olmesartan 40mg qd, furosemide 40mg bid at Nephrology department during on peritoneal dialysis. Post-PCI medication added from Cardiology department were clopidogrel 75mg qd, Rosuvastatin 10mg / ezetimibe 10mg qd. We added the medication information in Case report part.

 For having been already prescribed for aspirin 100mg, Olmesartan 40mg, furosemide 80mg per day at Nephrology department, the operator additionally prescribed clopidogrel 75mg, a combination pill of rosuvastatin 10mg / ezetimibe 10mg, bisoprolol 2.5mg, and nicorandil 10mg per day.  (Page 3, line 100 ~ 103)

- In Conclusions - the first two sentences are rather not conclusions.

 Thank you for your comment. We agree that previous sentences more fit to summary than conclusions. We changed Conclusions part to deliver our message clearer as below.

Although present consensus suggests the benefit of PCI in stable coronary disease is limited to symptom relieving, some prospective studies suggest otherwise. The present case demonstrated that some high-risk lesion characteristics can be overlooked with CAG alone. Intravascular imaging techniques such as NIRS-IVUS may be considered in specific situations, for PCI in lesions with high-impact characteristics may minimize future event risk. (Page 4, line 134 ~ page 5, line 139)

Best regards

References

Prati, Francesco, Enrico Romagnoli, Laura Gatto, Alessio la Manna, Francesco Burzotta, Yukio Ozaki, Valeria Marco, et al. 2019. “Relationship between Coronary Plaque Morphology of the Left Anterior Descending Artery and 12 Months Clinical Outcome: The CLIMA Study.” European Heart Journal, August. https://doi.org/10.1093/eurheartj/ehz520.

Waksman, Ron, Carlo di Mario, Rebecca Torguson, Ziad A Ali, Varinder Singh, William H Skinner, Andre K Artis, et al. 2019. “Identification of Patients and Plaques Vulnerable to Future Coronary Events with Near-Infrared Spectroscopy Intravascular Ultrasound Imaging: A Prospective, Cohort Study.” The Lancet 394 (10209): 1629–37. https://doi.org/10.1016/S0140-6736(19)31794-5.

Reviewer 2 Report

In this case report Kim et al showed the case of a 58-y old man with CKD admitted for exertion dyspnea. As the CAG showed a mild lesion in the mid LAD they performed NIRS-IVUS revealing a high plaque burden in the lesion. Although the case shown is interesting I have several concerns regarding the case:

1. Why the author did not perform any FFR. As there was no evidence of ischemia, the plaque burden is not a sufficient indication to perform PCI. 

2. Where other causes of exertion dyspnea excluded? I.e. obesity, high pulmonary pressures, etc. 

3. What was the medical therapy of the patient? Was the medical therapy optimised before proceeding with a CAG?

I think that, despite the interesting NIRS-IVUS finding, there was a weak indication to perform a PCI.

Author Response

Reviewer 2

In this case report Kim et al showed the case of a 58-y old man with CKD admitted for exertion dyspnea. As the CAG showed a mild lesion in the mid LAD they performed NIRS-IVUS revealing a high plaque burden in the lesion. Although the case shown is interesting I have several concerns regarding the case:

  1. Why the author did not perform any FFR. As there was no evidence of ischemia, the plaque burden is not a sufficient indication to perform PCI.

We thank the Reviewer for the critical comment. As the lesion was in clinically important location and showing inconsistent contrast density, the operator decided to check morphologic plaque characteristics, for they possibly imply future event risk. Physiologic study as FFR was not performed for it may result additional economic burden to the patient. We elaborated the reason for choosing NIRS-IVUS in Case report part.

 The lesion was located relatively proximal part of LAD and the contrast density within the lesion was inconsistent. The operator decided to evaluate high-risk plaque characteristics with NIRS-IVUS, for they may imply future event risk. Additional physiologic study was not performed for economic reason. (Page 2, line 51 ~ 54)

  1. Where other causes of exertion dyspnea excluded? I.e. obesity, high pulmonary pressures, etc.

 We thank you for your questions. We believe your issues are partially overlapped with those of comments of Reviewer 1. Obesity could be the potential reason for the symptom, therefore dietary and body weight control was recommended. On the peri-procedural echocardiogram, there was no clinically significant tricuspid valve regurgitation or elevated right ventricular systolic pressure, thus pulmonary hypertension was ruled out. We revised regarding issues in the Case report part.

He was a never smoker. His height was 163.6cm and the body weight was 80.1kg. (Body-mass-index 30kg/m2) (Page 1, line 38 ~ 40)

At the time, NT-proBNP was 1797 pg/mL and echocardiogram of the patient demonstrated no left ventricular (LV) systolic, no significant valvular dysfunction, no pulmonary hypertension, and grade 1 diastolic dysfunction. (LV ejection fraction 65%; only trivial tricuspid regurgitation without elevated right ventricular systolic pressure), The potential causes for the symptom could be ESRD and obesity.  (Page 3, line 93 ~ 97)

  1. What was the medical therapy of the patient? Was the medical therapy optimised before proceeding with a CAG?

The medications for the patient before CAG by Nephrology department were aspirin 100mg, Olmesartan 40mg, furosemide 80mg per day. At the time the patient visited Cardiology department, the reason for dyspnea was the first to be checked before any additional medication. The operator considered him as a high-risk patient and decided to perform CAG, because the patient was on dialysis. Post-PCI medications added from Cardiology department were clopidogrel 75mg, a combination pill of rosuvastatin 10mg / ezetimibe 10mg, bisoprolol 2.5mg, and nicorandil 10mg per day. We added the medication information in Case report part.

 For having been already prescribed for aspirin 100mg, Olmesartan 40mg, furosemide 80mg per day at Nephrology department, the operator additionally prescribed clopidogrel 75mg, a combination pill of rosuvastatin 10mg / ezetimibe 10mg, bisoprolol 2.5mg, and nicorandil 10mg per day. (Page 3, line 100 ~ 103)

I think that, despite the interesting NIRS-IVUS finding, there was a weak indication to perform a PCI.

Thank you for your critical comment. The comment is closely related with the Editor’s one and our corresponding reply. Indication by current guideline(Neumann et al. 2019) did not fully support PCI situation of the presented case. However, the location risk may be equivalent to that of proximal LAD lesion. Approximated minimal diameter stenosis by IVUS profiles was 48%, which slightly didn’t meet diameter stenosis cutoff value, MaxLCBI4mm was 462. LRP study suggest Max LCBI greater than 400 indicate higher risk for subsequent cardiovascular event. (Waksman et al. 2019) CLIMA study demonstrated morphologic features in LAD can be associated with a higher risk of future coronary events. (Prati et al. 2019) We concluded that PCI may benefit prognosis of the patient. We elaborated PCI decision process as below, in Case report part.

 Minimum lumen area (MLA) was 3.7mm2 and percent area stenosis was 61.2%. This could be approximately translated into percent diameter stenosis of 48%, which slightly did not meet significant stenosis. Despite absolute guideline cutoff in diameter stenosis was not enough for percutaneous coronary intervention (PCI), high-lipid burden indicated by MaxLCBI4mm and LAD location factors may impact future cardiac event risk. [2,5] To maximally benefit the future prognosis of the patient, the operator decided to perform PCI. (Page 2, line 57 ~ 64)

References

Neumann, Franz-Josef, Miguel Sousa-Uva, Anders Ahlsson, Fernando Alfonso, Adrian P Banning, Umberto Benedetto, Robert A Byrne, et al. 2019. “2018 ESC/EACTS Guidelines on Myocardial Revascularization.” European Heart Journal 40 (2): 87–165. https://doi.org/10.1093/eurheartj/ehy394.

Prati, Francesco, Enrico Romagnoli, Laura Gatto, Alessio la Manna, Francesco Burzotta, Yukio Ozaki, Valeria Marco, et al. 2019. “Relationship between Coronary Plaque Morphology of the Left Anterior Descending Artery and 12 Months Clinical Outcome: The CLIMA Study.” European Heart Journal, August. https://doi.org/10.1093/eurheartj/ehz520.

Waksman, Ron, Carlo di Mario, Rebecca Torguson, Ziad A Ali, Varinder Singh, William H Skinner, Andre K Artis, et al. 2019. “Identification of Patients and Plaques Vulnerable to Future Coronary Events with Near-Infrared Spectroscopy Intravascular Ultrasound Imaging: A Prospective, Cohort Study.” The Lancet 394 (10209): 1629–37. https://doi.org/10.1016/S0140-6736(19)31794-5.

Reviewer 3 Report

Very well written paper. 

Probably larger studies in the future can direct us on how to handle such patients as described in this case report

Author Response

Reviewer 3

Very well written paper.

Probably larger studies in the future can direct us on how to handle such patients as described in this case report

We thank you and agree with your comment. Larger studies with prospective design are mandatory to clearly understand clinical high-impact plaque characteristics, related with patient prognosis.

Reviewer 4 Report

The authors describe a case of non-obstructive CAD with high LCBI that was treated with DES in a patient with dyspnea.

While the technology of IVUS-NIRS is important in showing non-obstructive vulnerable plaque areas, it is still unknown whether treating such lesions is beneficial. Therefore it is debatable whether we should intervene or leave the patient to medical therapy alone.

Author Response

Reviewer 4

The authors describe a case of non-obstructive CAD with high LCBI that was treated with DES in a patient with dyspnea.

While the technology of IVUS-NIRS is important in showing non-obstructive vulnerable plaque areas, it is still unknown whether treating such lesions is beneficial. Therefore it is debatable whether we should intervene or leave the patient to medical therapy alone.

 We agree with the Reviewer that risk-benefit of intervention at a plaque with high-risk morphology in stable patients is unclear. Until concrete consensus is made, individualization based on specific clinical situation will be required to optimize patients’ treatment.

Round 2

Reviewer 4 Report

No further comments